# Advance care planning (ACP) to promote receipt of value-concordant care: Results vary according to patient priorities

**Holly G. Prigerson** *, **Martin Viola, Paul K. Maciejewski, Francesca Falzarano**

Cornell Center for Research on End-of Life Care, Department of Medicine, Weill Cornell Medicine, New York, New York, United States of America

* hgp2001@med.cornell.edu

## Abstract

### Background

Benefits of advance care planning (ACP) have recently been questioned by experts, but ACP is comprised of discrete activities. Little is known about which, if any, ACP activities are associated with patients' greater likelihood of receiving value-concordant end-of-life (EoL) care.

### Objectives

To determine which ACP activities [Do-Not-Resuscitate (DNR) order completion, designation of a healthcare proxy (HCP), and/or EoL discussions with physicians], individually and in combination, are associated with the greatest likelihood of receiving value-concordant care, and how results may vary based on patient-reported EoL care priorities.

### Methods

Data from 2 federally-funded, multisite, prospective cohort studies of EoL cancer care from 2002–2019 were analyzed. Cancer patients (N = 278) with metastatic disease refractory to chemotherapy were interviewed for a baseline assessment and followed prospectively until death. Interviews regarding patient priorities occurred a median of 111 days prior to death; data regarding EoL medical care were collected post-mortem from caregiver interviews and medical record abstraction. Patients who 1) prioritized life-extending care, and then received life-extending care (or avoided hospice care), or 2) prioritized comfort-focused care, and then avoided life-extending care (or received hospice care) in the last week of life, were coded as receiving value-concordant care.

### Results

After inverse propensity score weighting, the ACP combination associated with the largest proportion of patients receiving value-concordant care was DNR, HCP, and EoL discussions (87% vs. 64% for no ACP activities; OR = 3.91, p = 0.006). In weighted analyses examining each ACP activity individually, DNR orders were associated with decreased likelihood of

**Data Availability Statement:** All relevant data are within the paper.

**Funding:** : This research was supported, in part, by the following grants from the National Institute of

Health: R01 MH63892 (HGP) from the National Institute of Mental Health, R01 CA106370 (HGP) from the National Cancer Institute; R01 MD007652 (PKM, HGP) from the National Institute on Minority Health and Health Disparities; R35 CA197730 (HGP) from the National Cancer Institute; and UL1 TR002384 (Weill Cornell Medicine CTSC) from the National Center for Advancing Translational Sciences, from the National Institute on Aging-funded K99 grant (K99 AG073509). Dr. Prigerson, Dr. Maciejewski, and Dr. Falzarano have received NIH funding that supports some of their research effort The funders had no role in study design, data collection and analysis, decision to publish, or preparation of the manuscript.

**Competing interests:** The authors have declared that no competing interests exist.

life-extending care (89% vs. 75%; p = 0.005) and EoL discussions were associated with increased likelihood of hospice care (77% vs. 55%; p = 0.002) among patients prioritizing comfort. ACP activities were not significantly associated with increased likelihood of receiving value-concordant care among patients prioritizing life-extension.

## Conclusions and relevance

For patients who prioritize comfort, EoL discussions with physicians and completion of DNR orders may improve odds of receiving value-concordant EoL care. For patients who prioritize life-extension, ACP does not appear to improve odds of receiving value-concordant EoL care.

## Introduction

Advance care planning (ACP), including the completion of Do-Not-Resuscitate (DNR) orders and designation of a healthcare proxy (HCP), was devised to promote value-concordant care for dying patients lacking decisional capacity. Without explicit documentation otherwise, the default in most hospitals in the United States has been to administer life-prolonging care that physicians often consider overly burdensome and unbeneficial [1]. ACP gained momentum following the Patient Self Determination Act of 1990 [2–4] to provide patients the right to decide ahead of time the type of life-sustaining measures they would want, or not, should they become incapable of communicating their end-of-life (EoL) care wishes.

While ACP has been embraced by clinicians, patients, researchers [5–10] and the Centers for Medicare & Medicaid Services [11] as an indicator of high quality EoL care, it has also been derided by leaders in palliative medicine. Morrison et al. [12] have asserted that ACP is an ineffective instrument for ensuring that dying patients receive value-concordant care near death, citing 2 comprehensive reviews which failed to find evidence linking ACP to patients' receipt of goal-concordant EoL care [13, 14]. Similarly, in a 2022 *New York Times* essay, Lamas, a critical-care physician, questioned whether the current model of ACP works well enough to improve EoL care outcomes [15]. Other critical-care physicians and palliative care experts such as Curtis [5], have countered with examples, including quite poignantly his own, suggesting the comfort, preparation, and peace of mind that ACP affords. Consistent with this view, our own research has demonstrated that elements of ACP [16], such as engaging in EoL discussions [17, 18] and/or DNR order completion [19], are associated with patients' receipt of fewer unbeneficial procedures [16–19], less perceived suffering and loss of dignity [19], and higher rates of patients' receipt of value-concordant EoL care [18].

But perhaps not all aspects of ACP have equal impact on EoL outcomes. Surprisingly little is known of how elements of ACP, alone or in combination, relate to the odds that patients will receive EoL care aligned with their preferences. As noted above, using data from our Coping with Cancer 1 (CWC 1) study, we have found EoL discussions with physicians to be associated with patients' increased likelihood of receiving value-concordant EoL care [18], but we still do not know how these discussions compare to DNR order completion and/or designation of a healthcare proxy (HCP) in terms of their associations with patients' receipt of value-concordant EoL care. Identifying which ACP activities are most impactful, as well as the combinations of ACP activities associated with the highest rates of cancer patients' receipt of value-concordant care, could inform recommended strategies for obtaining desired EoL care.

There is also reason to believe that patient preferences and priorities not only influence the intensity of the care received near death [20], but also the likelihood of receipt of care consistent with those preferences [18]. For example, an analysis of 301 patients with advanced cancer recruited in CWC 1 found that the 27.6% who reported preferring life-extending therapy, regardless of the discomfort imposed by such care, were nearly 3 times more likely to receive intensive care and over 2 times less likely to receive hospice care than patients with a preference for EoL care focused on alleviation of pain and discomfort [20]. Another report using the same dataset found that 68% of patients received EoL care consistent with the care preferences that they had reported in a baseline assessment, but that rates varied dramatically based on the patient's EoL treatment preferences. For example, 59% of patients who stated that they preferred EoL care prioritizing relieving pain and discomfort then avoided life-extension treatment, but only 9% who prioritized life-prolonging treatment received it [18]. In light of these results, a significant question that remains is whether the ACP activities associated with receipt of value-concordant care would differ based on the patient's stated EoL care preferences.

Rather than lump ACP activities together as a single entity, our aim in this study is to examine the ACP activities associated with the greatest proportions of patients receiving care consistent with their reported EoL care preferences compared to not engaging in these ACP activities. Using a sample of 278 patients with advanced cancer recruited in CWC 1, as well as in a follow-up study, Coping with Cancer 3 (CWC 3), we sought to determine and compare the ACP activities, alone and in combination, associated with the highest likelihood of receipt of value-concordant EoL care. We then stratified analyses based on the patient's self-reported EoL care-related priorities. Based on prior findings [17–19], we expected that EoL discussions with physicians and DNR order completion would be associated with increased likelihood of patients receiving value-concordant EoL care compared to not engaging in any of the examined ACP activities. Consistent with prior results [18] we also hypothesized that ACP activities would have a greater effect on the receipt of value-concordant EoL care for patients with a stated preference for care prioritizing comfort compared to life-prolongation.

## Methods

### Sample

Study participants were enrolled in CWC 1 (CA106370) and CWC 3 (MD007652)– 2 multisite, prospective, longitudinal cohort studies of terminally ill cancer patients funded by the National Institutes of Health. Patients were eligible to participate in these studies if they were adults (at least 20 years old for CWC 1 and at least 21 years old for CWC 3) and had metastatic cancer refractory to 1st-line chemotherapy (or, for some CWC 3 participants with specific cancers, locally advanced cancer and/or disease progression on at least 2 lines of chemotherapy). Patients were ineligible to participate if they were not fluent in English or Spanish, displayed cognitive impairment as measured by a score of less than 6 on the Short Portable Mental Status Questionnaire, were unable to identify an informal caregiver, were enrolled in hospice or receiving palliative care, or were judged by interviewers or clinical staff as lacking stamina to participate in study activities.

Patients in CWC 1 were recruited from September 2002 through February 2008 from Yale Cancer Center (New Haven, CT), the Veterans Affairs Connecticut Healthcare System Comprehensive Cancer Clinics (West Haven, CT), Parkland Hospital (Dallas, TX), Simmons Comprehensive Cancer Center (Dallas, TX), Massachusetts General Hospital (Boston, MA), Dana-Farber Cancer Institute (Boston, MA), and New Hampshire Oncology-Hematology (Hookset, NH). Patients in CWC 3 were recruited from November 2015 through May 2019 from Memorial Sloan-Kettering Cancer Center (New York, NY), Columbia University Medical Center

(New York, NY), Northwestern University Robert H. Lurie Comprehensive Cancer Center (Chicago, IL), Rush University Medical Center (Chicago, IL), University of Texas-Southwestern (Dallas, TX), University of Texas at El Paso (El Paso, TX), and University of Miami Health System (Miami, FL). Institutional review board approval was obtained at all study sites, including Weill Cornell Medicine, which served as the single Institutional Review Board of record for CWC 3. Written informed consent was provided by all study participants.

CWC 1 and CWC 3 participants were included in this analysis if they died during the study period and had complete data on demographic characteristics, EoL treatment preferences, ACP, and life-extending care they received at the EoL. This produced a sample of 311, but 33 of these patients declined to choose between 2 care priorities, yielding an analytic sample of 278.

## Protocol

Data were collected via an in-person baseline interview with the patient and a post-mortem interview conducted with persons most familiar with the patient's care at that time. In CWC 3, post-mortem interviews were conducted with 1 of the patient's informal caregivers. In CWC 1, post-mortem interviews were conducted with either the patient's informal caregiver or medical staff (e.g., hospice nurse) who last cared for the patient in addition to abstracting relevant information from the patient's medical records. Patients in our sample died a median 111 days after baseline interview (IQR 59 to 216 days). Postmortem assessments occurred a median of 24 days after patient death (IQR 7 to 64 days).

## Measures

**Treatment preferences.** At baseline, patients were asked "if you could choose, would you prefer (1) a course of treatment that focused on extending life as much as possible, even if that meant more pain and discomfort, or (2) a plan of care that focused on relieving pain and discomfort, even if that meant not living as long?" As noted in other CWC analyses utilizing this methodology [18], this question was initially used in the Study to Understand Prognoses and Preferences for Outcomes and Risks of Treatments (SUPPORT trial) [21].

**EoL care.** Use of mechanical ventilation, cardiopulmonary resuscitation, tube feeding, chemotherapy, and/or treatment in an intensive care unit (ICU) during the patient's last week of life were collected through the post-mortem interview. Use of any of these interventions in the last week of life was used to create a dichotomous variable indexing life-extending care. Use of inpatient or outpatient hospice for at least one week was used to create a dichotomous variable indicating receipt of hospice care.

**Value-concordant EoL care.** Patients who reported a desire for life-extending care at baseline and who received life-extending care were coded as receiving value-concordant care, as were patients who reported a desire for care that prioritized comfort and who avoided life-extending care. Patients who were mismatched on baseline treatment preferences and EoL care were coded as not receiving value-concordant care–a method used in prior published analyses of CWC data [18]. This dichotomous variable served as the primary outcome. We also operationalized value-concordant care using hospice to indicate comfort-focused care at the EoL. Here patients who prioritized life-extending care and did not receive at least one week of hospice care, as well as patients who prioritized comfort-focused care and did receive at least one week of hospice care, were coded as receiving value-concordant care. Those who were mismatched on hospice utilization and baseline treatment preferences were coded as not receiving value-concordant care.

**ACP.** Patients reported at baseline interview whether they had (1) engaged in a discussion with their doctor about their EoL care priorities, (2) completed a DNR order, and (3) completed HCP/durable power of attorney documentation. We focused on these 3 elements because we consider them to be the most impactful and common; we excluded living wills as they were highly collinear with DNR order completion, are less common, and likely less impactful compared to DNR orders. These variables were used in 4 ways: they were examined independently, used to create an 8-level categorical variable of all their possible combinations, used to create a dichotomous variable of any versus no ACP, and used to create a dichotomous variable of all ACP elements at once versus no or any other ACP combination.

**Demographic data.** Patients' self-reported age, sex, race, and ethnicity were obtained in baseline interviews.

### Analysis

First, we described differences in demographic characteristics and treatment preferences among patients with any versus no ACP using frequencies, percentages, and chi-squared tests for categorical variables and medians, interquartile ranges, and Wilcoxon rank sum tests for continuous variables. We also compared the CWC 1 and CWC 3 cohorts on treatment preferences. Then, we used stabilized inverse propensity scores to weight the data and account for potential confounding due to sample characteristics. Weights were assigned using non-parametric covariate balancing propensity scores with age, sex, race, ethnicity, and EoL treatment preference used as predictors for a dichotomous variable of any versus no ACP. Standardized mean differences (SMD) were used to assess covariate balance before and after weighting, and a SMD <0.10 post-weighting was considered to represent acceptable balance between groups [22].

Next, we used weighted frequencies, percentages, and bivariate logistic regression models to compare the proportion of patients who received value-concordant care by 1) any versus no ACP and 2) all possible combinations of ACP elements. We then stratified our sample by EoL treatment preference and used weighted frequencies, percentages, and chi-square tests with Rao and Scott's correction to examine each ACP element separately and in aggregate, assessing the odds of receiving value-concordant care by type of ACP among each preference group. For each comparison in this stratified analysis, the data were reweighted using the same methodology as above. Lastly, we repeated our analyses using a definition of value-concordant care based on hospice utilization. All analyses were completed using R 4.1.1. Two-sided tests were used with p<0.05 taken to be statistically significant.

### Results

Demographic characteristics and treatment preferences of the sample used in this analysis can be found in Table 1. This sample included 243 participants from CWC 1 and 35 participants from CWC 3. Baseline treatment preferences between cohorts were similar; 183/243 (75%) of participants in CWC 1 and 26/35 (74%) of participants in CWC 3 expressed a preference for comfort-focused EoL care, with the remaining participants in each cohort expressing a preference for life-extending care. Following initial weighting, the sample was balanced at a SMD <0.10 level on all observed covariates (Fig 1).

Proportions of value-concordant care were similar among patients who engaged in any ACP compared to those who did not (71% versus 64%, OR = 1.37, 95% CI = 0.75–2.51, p = 0.303) (Table 2). The combination of ACP elements associated with the largest proportion of value-concordant care was DNR, HCP, and EoL discussions with physicians together (i.e., all possible ACP elements at once) (44/50, 87%) (Table 2). Patients who completed these 3 ACP elements were significantly more likely to receive value-concordant care than patients

**Table 1. Demographic characteristics and treatment preferences of unweighted sample (N = 278).**

| Characteristic | No ACP, N = 80[1] | Any ACP, N = 198[1] | p[2] |
|---|---|---|---|
| Age | 56 (47, 67) | 62 (54, 70) | 0.003 |
| Gender | | | 0.632 |
| Male | 40 (50%) | 107 (54%) | |
| Female | 40 (50%) | 91 (46%) | |
| Race | | | <0.001 |
| White | 34 (42%) | 139 (70%) | |
| Black | 18 (22%) | 36 (18%) | |
| Other | 28 (35%) | 23 (12%) | |
| Ethnicity | | | <0.001 |
| Not Hispanic | 54 (68%) | 174 (88%) | |
| Hispanic | 26 (32%) | 24 (12%) | |
| Treatment preferences | | | 0.083 |
| Extending life | 26 (32%) | 43 (22%) | |
| Relieving pain | 54 (68%) | 155 (78%) | |

[1]Median (IQR); n (%)

[2]Wilcoxon rank sum test; Pearson's Chi-squared test

Note: ACP = advance care planning

who did not engage in ACP (OR = 3.91, 95% CI = 1.50–10.2, p = 0.006). The combination of ACP elements associated with the smallest proportion of value-concordant care was HCP only (15/31, 50%).

For each comparison in Table 3, the sample was re-weighted and remained balanced at a SMD <0.10 level on all observed covariates. Among patients who desired comfort-focused care, those who discussed EoL care preferences with their physician more frequently received value-concordant care than those who did not (88% versus 74%, p = 0.013). Having a DNR was significantly associated with more frequent receipt of value concordant care among those who desired comfort-focused care (89% versus 75%, p = 0.005) but not among those who desired life-extending care (33% versus 27%, p = 0.666). Among patients who desired life-extending care, those who engaged in any form of ACP received value-concordant care almost as frequently as those who did not (30% versus 32%, p = 0.880). Utilizing all observed ACP elements was associated with increased frequency of value-concordant care, both among patients who prioritized comfort (95% versus 77%, p = 0.002) and in a small, non-statistically significant set of observations among those who prioritized extending life (30% versus 25%, p = 0.848).

In analyses examining value-concordant care as defined by hospice utilization (N = 275) (Note: 3 participants had missing data on hospice utilization), ACP element combinations that involved EoL discussions with physicians had the largest proportions of value-concordant care (Table 4). Among patients prioritizing comfort-focused care, EoL discussions and utilizing all ACP elements remained significantly associated with improved odds of value-concordant care, but DNR was not (Table 5). Among patients who prioritized life-extending care, those who did *not* engage in any ACP trended towards more frequently receiving value-concordant care (61% versus 37%, p = 0.073).

## Discussion

Morrison et al. [12] conclude that ACP is not associated with better EoL outcomes overall, and specifically with respect to increased rates of patient receipt of value-concordant care. But, to

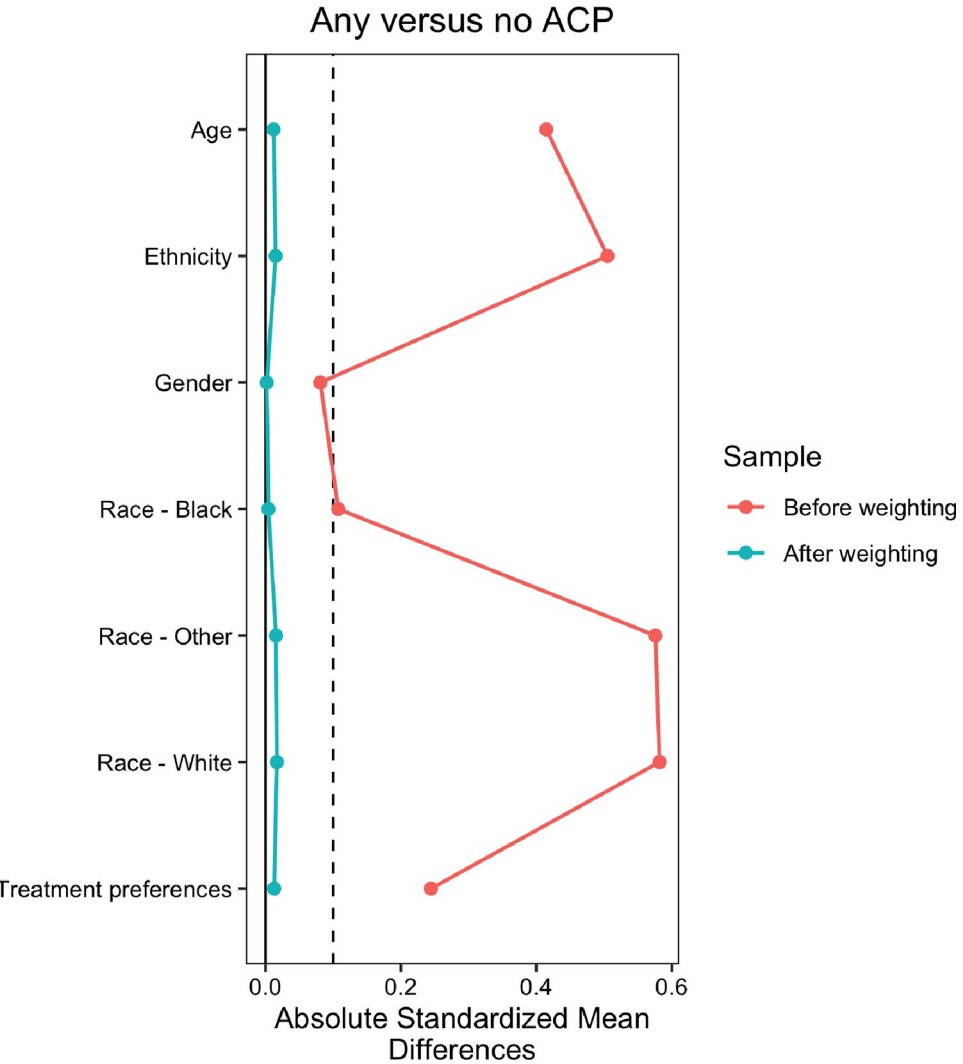

**Fig 1. Displays the absolute standard mean differences in sociodemographic characteristics of the sample depending on whether they had engaged in any of the examined aspects of advance care planning (ACP) or not.** On the left, green indicates the differences on sociodemographic characteristics after weighting. On the right, red indicates the differences before weighting. The figure shows that weighting effectively neutralized differences on these sociodemographic characteristics.

date, the debate about the effectiveness of ACP on EoL outcomes has lacked granularity. We here examined specific elements of ACP that may or may not, alone or in combination, promote receipt of value-concordant EoL care, and for whom, based on patients' self-reported EoL care preferences. These results suggest that there may be effective ways for patients to engage in ACP and receive value-concordant care if they prioritize comfort, but that ACP is not helpful to promoting value-concordant care for patients who prioritize life-prolongation.

Our results revealed that the combination of ACP elements most associated with EoL care consistent with patient preferences included DNR order completion, designation of an HCP, and engaging in EoL discussions with physicians. It is unsurprising that the more elements of ACP completed, the more likely patients will receive value-concordant care. Still, it appears remarkable that almost 90% of patients who engaged in EoL discussions, signed a DNR order, and assigned an HCP– 3 activities—received EoL care aligned with their baseline reported

**Table 2. Odds of receiving value-concordant care by type of ACP (N = 278).**

| Type of ACP | Weighted proportion receiving value-concordant care | OR | 95% CI | p |
|---|---|---|---|---|
| **Any versus no ACP** | | | | |
| No ACP | 51/80 (64%) | — | — | |
| Any ACP | 140/198 (71%) | 1.37 | 0.75, 2.51 | 0.303 |
| **By ACP element** | | | | |
| No ACP | 51/80 (64%) | — | — | |
| EoL discussions with physicians, DNR, and HCP | 44/50 (87%) | 3.91 | 1.50, 10.2 | 0.006 |
| EoL discussions with physicians and DNR | 14/18 (77%) | 1.93 | 0.55, 6.80 | 0.304 |
| DNR | 10/14 (76%) | 1.77 | 0.42, 7.38 | 0.435 |
| EoL discussions with physicians | 25/34 (74%) | 1.60 | 0.58, 4.44 | 0.363 |
| HCP and DNR | 19/31 (62%) | 0.95 | 0.39, 2.29 | 0.900 |
| EoL discussions with physicians and HCP | 12/20 (59%) | 0.83 | 0.30, 2.35 | 0.730 |
| HCP | 15/31 (50%) | 0.56 | 0.22, 1.41 | 0.219 |

Data weighted for any versus no ACP using stabilized inverse propensity score weights

Note: EoL = end of life, HCP = health care proxy, DNR = do-not-resuscitate order

care preferences. These results are consistent with our prior research suggesting that EoL discussions are associated with a greater likelihood of receiving value-concordant care [18]. As another example, prior research also found that DNR order completion within the first 48 hours of an ICU admission was associated with receipt of fewer aggressive medical interventions and less suffering, though not explicitly value-concordant care [19]. Taken together, our results add to the literature demonstrating that ACP activities are potentially effective tools for supporting patients' receipt of EoL care consistent with their values.

That said, all ACP activities may not be equally effective, and their efficacy may vary depending on patients' baseline reported preferences for EoL care. We learned that among the

**Table 3. Weighted proportion of patients receiving value-concordant care by treatment preference and type of advance care planning (N = 278).**

| | Extending life | | Relieving pain/discomfort | |
|---|---|---|---|---|
| Type of ACP | Weighted proportion receiving value-concordant care | p | Weighted proportion receiving value-concordant care | p |
| EoL discussions with physicians | | 0.993 | | 0.013 |
| No | 11/38 (28%) | | 87/118 (74%) | |
| Yes | 8/31 (28%) | | 80/91 (88%) | |
| DNR | | 0.666 | | 0.005 |
| No | 11/39 (27%) | | 90/121 (75%) | |
| Yes | 10/29 (33%) | | 79/89 (89%) | |
| HCP | | 0.722 | | 0.964 |
| No | 10/34 (29%) | | 84/104 (81%) | |
| Yes | 12/35 (34%) | | 85/105 (81%) | |
| Any ACP | | 0.880 | | 0.129 |
| No | 6/20 (32%) | | 45/60 (74%) | |
| Yes | 15/49 (30%) | | 125/149 (84%) | |
| All ACP | | 0.848 | | 0.002 |
| No | 14/55 (25%) | | 130/168 (77%) | |
| Yes | 4/14 (30%) | | 39/41 (95%) | |

Data reweighted using stabilized inverse propensity score weights for each comparison

**Table 4. Odds of receiving value-concordant care by type of ACP using hospice for at least one week as sole indicator of value-concordant care (N = 275).**

| Type of ACP | Weighted proportion receiving value-concordant care | OR[1] | 95% CI[1] | p |
|---|---|---|---|---|
| **Any versus no ACP** | | | | |
| No ACP | 44/79 (56%) | — | — | |
| Any ACP | 123/196 (63%) | 1.30 | 0.71, 2.38 | 0.386 |
| **By ACP element** | | | | |
| No ACP | 44/79 (56%) | — | — | |
| EoL discussions with physicians and DNR | 13/17 (78%) | 2.83 | 0.71, 11.3 | 0.140 |
| EoL discussions with physicians and HCP | 13/19 (69%) | 1.72 | 0.57, 5.19 | 0.333 |
| EoL discussions with physicians, DNR, and HCP | 34/50 (67%) | 1.57 | 0.72, 3.40 | 0.252 |
| EoL discussions with physicians | 22/34 (63%) | 1.34 | 0.52, 3.47 | 0.550 |
| HCP and DNR | 18/31 (59%) | 1.10 | 0.46, 2.63 | 0.831 |
| DNR | 8/14 (57%) | 1.05 | 0.29, 3.81 | 0.944 |
| HCP | 15/31 (49%) | 0.74 | 0.29, 1.86 | 0.521 |

[1]OR = Odds Ratio, CI = Confidence Interval

Data weighted for any versus no ACP using stabilized inverse propensity score weights

studied advanced cancer patients who prioritized comfort-focused care at the EoL (which was the clear majority, 67%), engaging in EoL discussions with their physician and completion of DNR orders were both significantly associated with receipt of value-concordant EoL care. For those who engaged in all 3 of the examined ACP activities, 95% received value-concordant care. Further, when hospice care was used to indicate receipt of comfort-focused care at EoL, we found that EoL discussions with physicians was the sole individual ACP element significantly associated with receipt of value-concordant care among patients prioritizing comfort at the EoL. Thus, our results suggest that while there may be multiple pathways to avoiding intensive EoL care through ACP, EoL discussions with physicians may have unique potential for achieving hospice utilization among those prioritizing the alleviation of discomfort near death.

**Table 5. Weighted proportion of patients receiving value-concordant care by treatment preference and type of advance care planning, using hospice for at least one week as sole indicator of value-concordant care (N = 275).**

| | Extending life | | Relieving pain/discomfort | |
|---|---|---|---|---|
| Type of ACP | Weighted proportion receiving value-concordant care | p | Weighted proportion receiving value-concordant care | p |
| EoL discussions with physicians | | 0.062 | | 0.002 |
| No | 21/38 (54%) | | 64/117 (55%) | |
| Yes | 9/31 (28%) | | 69/89 (77%) | |
| DNR | | 0.274 | | 0.120 |
| No | 20/39 (52%) | | 72/118 (61%) | |
| Yes | 11/29 (36%) | | 63/88 (72%) | |
| HCP | | 0.267 | | 0.590 |
| No | 18/34 (51%) | | 69/101 (68%) | |
| Yes | 12/35 (35%) | | 74/104 (71%) | |
| Any ACP | | 0.073 | | 0.053 |
| No | 12/20 (61%) | | 32/59 (55%) | |
| Yes | 18/49 (37%) | | 105/147 (71%) | |
| All ACP | | 0.035 | | 0.021 |
| No | 27/55 (49%) | | 105/165 (64%) | |
| Yes | 0/14 (0%) | | 34/41 (83%) | |

Importantly, we also found that for advanced cancer patients who indicated that they would prioritize life-extension to comfort at the EoL, no individual ACP activity had a significant association with odds that they would get the life-extending care that they expressed preferring in the baseline interview. It appears that the healthcare system is structured to preserve and prolong life as the default, and that patients who desire that form of EoL care will likely get it without any ACP intervention. If anything, for those wishing to avoid hospice care, ACP may prove counterproductive. That said, from the perspective of a family surrogate decision-maker, engaging in EoL discussions to clarify the patient's prioritization of life-extending care may well promote the patient's receipt of that type of EoL care in the not uncommon circumstance in which the patient becomes incapacitated prior to death.

While we had expected HCP designation to be influential, our results suggest that HCPs alone are not effective for ensuring that patients who prioritize comfort will receive value-consistent EoL care. There are several potential explanations for this finding, including that patients and their HCPs may have conflicting preferences regarding EoL care and that HCPs do not necessarily ensure that the patient's, rather than their own, preferences will be honored. It may also be true that HCPs are unaware of the patient's priorities and/or that patient priorities may change from what they told the HCP and/or that the medical team may not be influenced by the HCP if they believe the HCP is not well-informed or acting in the best interest of the patient. Regardless of the reason, these results suggest that patients, HCPs, and oncologists should engage in EoL discussions so that all may be informed of the benefits and harms of the treatment options being presented to them and work to realize that the patient's *informed* wishes and priorities are respected.

Our study findings should be considered in the context of its limitations. Although our sample size was sufficient to demonstrate statistically significant differences in achieving value-concordant care, a larger sample would afford greater statistical power to test for subgroup effects. In addition, the EoL discussions only documented discussions between patients and their physicians, and not with other member of the care team (e.g., advance practice providers who can also bill for these discussions) or their family surrogate decision-maker. Future research should replicate these findings in larger and more diverse patient samples, including non-cancer patients, and inquire about EoL discussions with other clinical staff and family members. It is important to acknowledge that a key limitation of the present analysis is that EoL care priorities reported in a baseline interview may well have changed as patients approached death and thus our analysis may have misclassified instances of goal-concordant care. Additionally, our measure of EoL care with which we matched the baseline EoL care preferences was limited to care received in the final week of life. Future research should examine the effectiveness of other elements of ACP (e.g., living wills), patient care preferences closer to death, as well as include care received in the weeks leading up to the patient's final week. This report did not examine the role of trusting and empathic relationships with a professional/s which has been shown to be a key motivator for engaging individuals and families in ACP discussions [23–25]. Lastly, it is important to acknowledge that much of the sample was assessed in a study conducted in well over a decade ago. Given the changing landscape of ACP, the results may no longer hold and need to be confirmed in a more recently conducted study.

## Conclusion

Data from two prospective, longitudinal studies were used to examine which elements of ACP were associated with the greatest likelihood of receiving value-concordant EoL care. We found that for patients who prioritized comfort, EoL discussions with physicians and completion of

DNR orders were associated with significantly greater odds of receiving value-concordant EoL care. For patients who prioritized life-extension, ACP did not appear to improve odds of receiving value-concordant EoL care and may even reduce them. Results suggest the most effective forms of ACP to promote value-concordant EoL care based on patient priorities.

## Author Contributions

**Conceptualization:** Holly G. Prigerson, Martin Viola, Francesca Falzarano.

**Data curation:** Holly G. Prigerson.

**Formal analysis:** Martin Viola, Paul K. Maciejewski.

**Funding acquisition:** Holly G. Prigerson, Paul K. Maciejewski.

**Investigation:** Holly G. Prigerson, Paul K. Maciejewski, Francesca Falzarano.

**Methodology:** Holly G. Prigerson, Martin Viola, Paul K. Maciejewski.

**Project administration:** Holly G. Prigerson, Paul K. Maciejewski, Francesca Falzarano.

**Resources:** Holly G. Prigerson, Paul K. Maciejewski.

**Software:** Holly G. Prigerson, Martin Viola.

**Supervision:** Holly G. Prigerson, Paul K. Maciejewski, Francesca Falzarano.

**Validation:** Holly G. Prigerson, Martin Viola, Paul K. Maciejewski, Francesca Falzarano.

**Visualization:** Martin Viola, Paul K. Maciejewski.

**Writing – original draft:** Holly G. Prigerson, Martin Viola, Francesca Falzarano.

**Writing – review & editing:** Holly G. Prigerson, Martin Viola, Francesca Falzarano.

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
