## [Decision Letter · Decision Letter 0]

20 Sep 2022

PONE-D-22-15436Advance Care Planning to Promote Cancer Patients’ Receipt of Value-Concordant CarePLOS ONE

Dear Professor Prigerson,

Thank you for submitting your manuscript to PLOS ONE. After careful consideration, we feel that it has merit but does not fully meet PLOS ONE’s publication criteria as it currently stands. Therefore, we invite you to submit a revised version of the manuscript that addresses the points raised during the review process.

The manuscript has been evaluated by two reviewers, and their comments are available below. I, as Academic Editor, acted as one of the reviewers as this topic lies within my area of research and expertise. 

The reviewers have raised a number of concerns that need attention largely around interpretations of this study's findings and communicating its limitations. 

Could you please revise the manuscript to carefully address the concerns raised?

We look forward to receiving your revised manuscript.

Kind regards,

Cari Malcolm

Academic Editor

PLOS ONE

Journal Requirements:

2. In ethics statement in the manuscript and in the online submission form, please provide additional information about the patient records/samples used in your retrospective study. Specifically, please ensure that you have discussed whether all data/samples were fully anonymized before you accessed them and/or whether the IRB or ethics committee waived the requirement for informed consent. If patients provided informed written consent to have data/samples from their medical records used in research, please include this information.

This research was supported, in part, by the following grants from the National Institute of Health: R01 MH63892 (HGP) from the National Institute of Mental Health, R01 CA106370 (HGP) from the National Cancer Institute; R01 MD007652 (PKM, HGP) from the National Institute on Minority Health and Health Disparities; R35 CA197730 (HGP) from the National Cancer Institute; and UL1 TR002384 (Weill Cornell Medicine CTSC) from the National Center for Advancing Translational Sciences, from the National Institute on Aging-funded K99 grant (K99 AG073509).

NO The funders had no role in study design, data collection and analysis, decision to publish, or preparation of the manuscript.

7. Please include your tables as part of your main manuscript and remove the individual files. Please note that supplementary tables (should remain/ be uploaded) as separate "supporting information" files

Reviewers' comments:

Reviewer's Responses to Questions

**Comments to the Author**

1. Is the manuscript technically sound, and do the data support the conclusions?

Reviewer #1: Yes

Reviewer #2: Yes

2. Has the statistical analysis been performed appropriately and rigorously? 

Reviewer #1: I Don't Know

Reviewer #2: Yes

3. Have the authors made all data underlying the findings in their manuscript fully available?

Reviewer #1: Yes

Reviewer #2: Yes

4. Is the manuscript presented in an intelligible fashion and written in standard English?

Reviewer #1: Yes

Reviewer #2: Yes

5. Review Comments to the Author

Reviewer #1: This is an important analysis of that includes consideration of patient preferences and assesses goal concordance with outcomes, offering a more granular look at the influence of ACP and its included elements.

The manuscript is well-written and also speaks to the current controversy over the value and utility of ACP. There are some areas of clarity for consideration included below. Further, while the statistical methods appear appropriate, it would be best for a statistician to review the specific analyses described.

The authors include 3 components of ACP – EoL discussions with a physician, completion of a DNR, and identification of a proxy. There are some potential limitations with this structure that should be addressed. First, if this only includes EoL discussions with a physician, there may be important EoL discussions with other qualified clinical care providers that are missed (for example, advanced practice practitioners can also bill for ACP). But more importantly, are the three categories truly separate and without overlap- for example, is completing a DNR truly distinct from EoL discussion? It seems reasonable that some of these discussions might include a review of DNR status and decision-making, as well as who a patient might select as a proxy.

From the original study protocol- why exclude patients with terminal cancer if receiving palliative care? Wouldn’t this be clinically appropriate for many of these patients?

This paper does briefly note an important limitation in the discussion – the analysis assumes patient preferences remain stable from baseline measure to outcomes to assess goal concordance – and these preferences could change after the median 111 days.

It appears, although is not completely clear, that all ACP measures are exclusively collected at the baseline interview. Could a participant have engaged in or changed any of the elements noted at baseline prior to death that are not captured in the data (or indicated ‘no DNR’ at baseline but later had one? Is it possible to check for, for ex., updated DNR status along with the outcome data?) Reviews of EoL preference stability shows a range of variability across studies, although generally are more stable for seriously ill patients.

It would be helpful to include more discussion of implications for patients prioritizing life extending care. What are recommendations for this group?

Reviewer #2: This is a very well written paper communicating novel research around the complex issue of promoting value-concordant care in people with cancer through the use of advance care planning (ACP).

The introduction/background section is excellent and, importantly, highlights recent controversies and debate around the utility of and challenges around ACP. Key and recent evidence is integrated throughout this section, leading to the authors’ hypotheses.

The methods are detailed and clearly communicated.

Given that data/participants from two studies (CWC1 and CWC3) and therefore time periods (CWC1 2002-2008 and CWC2 2015-2019) were used in this retrospective analysis and cognisant of the fact that the landscape of ACP and value-concordant care has changed and evolved considerably over the nearly 20 years since 2002, this needs to be acknowledged by the authors and explicitly discussed in the limitations section. I would also like to see greater transparency around how comparable data from CWC1 and CWC2 participants were in terms of, particularly, treatment preferences. The 278 participants are communicated as a whole sample in table 1 so it isn’t clear.

An additional limitation which needs to be addressed is around the inclusion of solely end of life care discussions with a physician. It is more than often the case that there is overlap and ongoing supportive discussions from a range of professionals, the family/significant others and the patient that lead to the patient’s decision around treatment preferences. There is also a growing body of evidence that having trusting and empathic relationships with a professional/s is a key motivator for engaging individuals and families in ACP discussions. The current study isn’t able to tease that out and therefore needs to acknowledge that. It may be worth referring to the following recent article in the discussion:

Rosa WE, Izumi S, Sullivan D, Lakin J, Rosenberg AR, Creutzfeldt CJ, Lafond D et al. Advance Care Planning in Serious Illness: A Narrative Review. Journal of Pain and Symptom Management. 2022 Aug 24 https://doi.org/10.1016/j.jpainsymman.2022.08.012

Finally, the fact that retrospective analysis is based around the patient’s preferences for end of life care to stay the same in the last weeks and months of life is a considerable limitation of this research and the findings and conclusions need to be interpreted with some caution. There needs to be greater transparency around this and it needs to be explicitly stated.

6. PLOS authors have the option to publish the peer review history of their article (what does this mean?). If published, this will include your full peer review and any attached files.

Reviewer #1: No

Reviewer #2: No

---

## [Author Response · Author response to Decision Letter 0]

1 Dec 2022

Comment #1: 

Journal Requirements:

https://journals.plos.org/plosone

/s/file?id=ba62/PLOSOne_formatting_sample_title_authors_affiliations.pdf

Response: We have made our manuscript compliant with PLOS ONE formatting.

2. In ethics statement in the manuscript and in the online submission form, please provide additional information about the patient records/samples used in your retrospective study. Specifically, please ensure that you have discussed whether all data/samples were fully anonymized before you accessed them and/or whether the IRB or ethics committee waived the requirement for informed consent. If patients provided informed written consent to have data/samples from their medical records used in research, please include this information.

Response: As stated in the manuscript, we obtained written informed consent from all study participants to have their data, including from their medical records, used in the study. 

The following is now input into the online Ethics Statement: All participating institutions obtained IRB approval; Weill Cornell Medicine protocol #1312014603 was the IRB of record. Written informed consent was obtained from all study participants to have their data, including from their medical records, used in the study. The analysis for this specific study did not include data from the patient’s medical chart but rather from the postmortem interviews with the caregiver (professional or family member) who was most familiar with the care the patient had received in the last week of life. Data from the medical chart were used only as a validity check on the postmortem entries.

Response: We apologize for any inconsistencies. The grant information and funding information and financial disclosures sections should all match now.

This research was supported, in part, by the following grants from the National Institute of Health: R01 MH63892 (HGP) from the National Institute of Mental Health, R01 CA106370 (HGP) from the National Cancer Institute; R01 MD007652 (PKM, HGP) from the National Institute on Minority Health and Health Disparities; R35 CA197730 (HGP) from the National Cancer Institute; and UL1 TR002384 (Weill Cornell Medicine CTSC) from the National Center for Advancing Translational Sciences, from the National Institute on Aging-funded K99 grant (K99 AG073509).

Response: We have removed funding information from the Acknowledgments Section, which itself has been deleted.

NO The funders had no role in study design, data collection and analysis, decision to publish, or preparation of the manuscript.

Response: The following USA federal agencies supported the data collection and effort of the authors: R01 MH63892 (HGP) from the National Institute of Mental Health, R01 CA106370 (HGP) from the National Cancer Institute; R01 MD007652 (PKM, HGP) from the National Institute on Minority Health and Health Disparities; R35 CA197730 (HGP) from the National Cancer Institute; and UL1 TR002384 (Weill Cornell Medicine CTSC) from the National Center for Advancing Translational Sciences, from the National Institute on Aging-funded K99 grant (K99 AG073509).

While these National Institute of Health agencies financially supported the study, they did not influence the study design, data collection, analysis, interpretation of results or preparation of this or any other manuscript from these data and there is no apparent or real conflict of interest. 

Response: We will make the data used in this report available once we have published the data testing the hypotheses in the specified aims of the study. We will not provide these data prior to publication of the awarded study’s specific aims.

Response: We have stated that we obtained informed written consent for study participation. We now add the name of the IRB that reviewed and approved the CWC 3 was the Weill Cornell Medicine IRB. Weill Cornell Medicine was the IRB of record and required each participating site to obtain local IRB approval. All participating study sites obtained local IRB approval. 

7. Please include your tables as part of your main manuscript and remove the individual files. Please note that supplementary tables (should remain/ be uploaded) as separate "supporting information" files

Response: Tables and Figure 1 are now included within the main manuscript and not as separate files.

Response: We have added references recommended by the reviewers.

Reviewers' comments:

Reviewer's Responses to Questions

Comments to the Author

1. Is the manuscript technically sound, and do the data support the conclusions?

Reviewer #1: Yes

Reviewer #2: Yes

2. Has the statistical analysis been performed appropriately and rigorously? 

Reviewer #1: I Don't Know

Reviewer #2: Yes

3. Have the authors made all data underlying the findings in their manuscript fully available?

Reviewer #1: Yes

Reviewer #2: Yes

4. Is the manuscript presented in an intelligible fashion and written in standard English?

Reviewer #1: Yes

Reviewer #2: Yes

5. Review Comments to the Author

Reviewer #1: This is an important analysis of that includes consideration of patient preferences and assesses goal concordance with outcomes, offering a more granular look at the influence of ACP and its included elements.

Response: We thank the reviewer for acknowledging that this analysis addresses an important, currently hotly contested debate regarding the value of ACP with respect to end-of-life outcomes.

The manuscript is well-written and also speaks to the current controversy over the value and utility of ACP. There are some areas of clarity for consideration included below. Further, while the statistical methods appear appropriate, it would be best for a statistician to review the specific analyses described.

The authors include 3 components of ACP – EoL discussions with a physician, completion of a DNR, and identification of a proxy. There are some potential limitations with this structure that should be addressed. First, if this only includes EoL discussions with a physician, there may be important EoL discussions with other qualified clinical care providers that are missed (for example, advanced practice practitioners can also bill for ACP). But more importantly, are the three categories truly separate and without overlap- for example, is completing a DNR truly distinct from EoL discussion? It seems reasonable that some of these discussions might include a review of DNR status and decision-making, as well as who a patient might select as a proxy.

Response: Thanks to the reviewer for noting the need to distinguish between patients’ discussions of the care they would want to receive if they were dying with their physician from other members of the care team who may also bill for these discussions (e.g., advance practice nurses). This is now acknowledged in the limitations section of the revised manuscript. With respect to overlap among the ACP components, it is worth noting that each aspect of ACP was assessed separately (e.g., DNR order completion, EoL discussions were coded separately) based on the patient’s perceptions and recall of what was discussed and documented. Future research that audio and/or video graphically records what was discussed and what was documented would help to distinguish between these components more objectively – a point we now note in the limitations section of the revised manuscript.

From the original study protocol- why exclude patients with terminal cancer if receiving palliative care? Wouldn’t this be clinically appropriate for many of these patients?

Response: We excluded advanced cancer patients who were receiving palliative care because that was an outcome we were predicting (i.e., we wanted to examine prospectively the factors that led to receipt of palliative care).

This paper does briefly note an important limitation in the discussion – the analysis assumes patient preferences remain stable from baseline measure to outcomes to assess goal concordance – and these preferences could change after the median 111 days.

It appears, although is not completely clear, that all ACP measures are exclusively collected at the baseline interview. Could a participant have engaged in or changed any of the elements noted at baseline prior to death that are not captured in the data (or indicated ‘no DNR’ at baseline but later had one? Is it possible to check for, for ex., updated DNR status along with the outcome data?) Reviews of EoL preference stability shows a range of variability across studies, although generally are more stable for seriously ill patients.

Response: As the reviewer notes, we acknowledge that patient preferences and ACP was assessed at baseline and that preferences and ACP activities could have occurred following that assessment. Results show, however, that regardless of what happened subsequent to the baseline assessment, the baseline assessed ACP activities did prove associated with EoL care received near death, highlighting that there was some influence regardless. It would be extraordinarily difficult to capture information on the interregnum for these deceased study participants at this point in time.

It would be helpful to include more discussion of implications for patients prioritizing life extending care. What are recommendations for this group?

Response: Results indicated that for patients prioritizing life-extending care there is less of a need for them to do anything as the system appears to be designed to provide life-extending care as the default. That said, it may be helpful for patients to engage in EoL discussions with the family surrogate decision-maker, who would ultimately make decisions regarding life-support should the patient become incapacitated. We now state this as follows in the revised Discussion:

“from the perspective of a family surrogate decision-maker, engaging in EoL discussions to clarify the patient’s preference for life-extending care may well promote the patient’s receipt of that type of EoL care in the not uncommon circumstance in which the patient becomes incapacitated prior to death.”

Reviewer #2: This is a very well written paper communicating novel research around the complex issue of promoting value-concordant care in people with cancer through the use of advance care planning (ACP).

The introduction/background section is excellent and, importantly, highlights recent controversies and debate around the utility of and challenges around ACP. Key and recent evidence is integrated throughout this section, leading to the authors’ hypotheses.

The methods are detailed and clearly communicated.

Given that data/participants from two studies (CWC1 and CWC3) and therefore time periods (CWC1 2002-2008 and CWC2 2015-2019) were used in this retrospective analysis and cognisant of the fact that the landscape of ACP and value-concordant care has changed and evolved considerably over the nearly 20 years since 2002, this needs to be acknowledged by the authors and explicitly discussed in the limitations section. 

Response: We thank the reviewer for the appreciation of our work. We do wish to clarify that the analysis is prospective (i.e., the baseline assessment is associated with subsequent outcomes retrospectively reported by a caregiver in the postmortem assessment). We now note in the limitations section that some, but not all, of the data were obtained from a study conducted nearly 20 years ago and that given the changing landscape of ACP the results obtained may no longer hold and should be confirmed in a more recently conducted study.

I would also like to see greater transparency around how comparable data from CWC1 and CWC2 participants were in terms of, particularly, treatment preferences. The 278 participants are communicated as a whole sample in table 1 so it isn’t clear.

Response: Thank you for this comment. In the Results section, we now report the number of participants from each study cohort and treatment preferences by CWC 1 vs. CWC 3 as follows:

“This sample included 243 participants from CWC 1 and 35 participants from CWC 3. Baseline treatment preferences between cohorts were similar; 183/243 (75%) of participants in CWC 1 and 26/35 (74%) of participants in CWC 3 expressed a preference for comfort-focused EoL care, with the remaining participants in each cohort expressing a preference for life-extending care.”

An additional limitation which needs to be addressed is around the inclusion of solely end of life care discussions with a physician. It is more than often the case that there is overlap and ongoing supportive discussions from a range of professionals, the family/significant others and the patient that lead to the patient’s decision around treatment preferences. There is also a growing body of evidence that having trusting and empathic relationships with a professional/s is a key motivator for engaging individuals and families in ACP discussions. The current study isn’t able to tease that out and therefore needs to acknowledge that. It may be worth referring to the following recent article in the discussion:

Rosa WE, Izumi S, Sullivan D, Lakin J, Rosenberg AR, Creutzfeldt CJ, Lafond D et al. Advance Care Planning in Serious Illness: A Narrative Review. Journal of Pain and Symptom Management. 2022 Aug 24 https://doi.org/10.1016/j.jpainsymman.2022.08.012

Response: This is an excellent point. As noted in response to Reviewer 1, we have added the need to acknowledge that there may be important discussions with other, non-physician, members of the care team. We also have added a suggestion of the potential benefits of engaging in EoL discussions with a family surrogate. We now acknowledge the role of therapeutic bond with a member of the healthcare team as facilitating and supporting EoL discussions, citing Rosa et al., but also our prior research that highlights the influential role of the therapeutic alliance with a member of the care team in promoting ACP discussions. 

Finally, the fact that retrospective analysis is based around the patient’s preferences for end of life care to stay the same in the last weeks and months of life is a considerable limitation of this research and the findings and conclusions need to be interpreted with some caution. There needs to be greater transparency around this and it needs to be explicitly stated.

Response: To clarify, the analysis was prospective and the baseline assessment of preferences were examined in relation to a subsequent (retrospective) postmortem assessment of the EoL care received. We have acknowledged that preferences may change as a limitation of this analysis. We now emphasize this as a key limitation.

6. PLOS authors have the option to publish the peer review history of their article (what does this mean?). If published, this will include your full peer review and any attached files.

Response: Publishing the review is fine with us.

---

## [Decision Letter · Decision Letter 1]

22 Dec 2022

Advance Care Planning (ACP) to promote receipt of value-concordant care: Results vary according to patient priorities

PONE-D-22-15436R1

Dear Dr. Prigerson,

We’re pleased to inform you that your manuscript has been judged scientifically suitable for publication and will be formally accepted for publication once it meets all outstanding technical requirements.

Kind regards,

Cari Malcolm

Academic Editor

PLOS ONE

Reviewer's Responses to Questions

**Comments to the Author**

1. If the authors have adequately addressed your comments raised in a previous round of review and you feel that this manuscript is now acceptable for publication, you may indicate that here to bypass the “Comments to the Author” section, enter your conflict of interest statement in the “Confidential to Editor” section, and submit your "Accept" recommendation.

Reviewer #1: All comments have been addressed

Reviewer #2: All comments have been addressed

2. Is the manuscript technically sound, and do the data support the conclusions?

Reviewer #1: Yes

Reviewer #2: Yes

3. Has the statistical analysis been performed appropriately and rigorously? 

Reviewer #1: I Don't Know

Reviewer #2: Yes

4. Have the authors made all data underlying the findings in their manuscript fully available?

Reviewer #1: Yes

Reviewer #2: Yes

5. Is the manuscript presented in an intelligible fashion and written in standard English?

Reviewer #1: Yes

Reviewer #2: Yes

6. Review Comments to the Author

Reviewer #1: a very minor final revision prior to publication - the revised section of the final paragraph in the discussion "not with other member of the care team" should state "not with other members of the care team"

Reviewer #2: Thank you for submitting your revised manuscript. I am satisfied that all recommendations have been considered and the current version is suitable for publication.

7. PLOS authors have the option to publish the peer review history of their article (what does this mean?). If published, this will include your full peer review and any attached files.

Reviewer #1: **Yes: **Amanda Jane Reich

Reviewer #2: No

---

## [Editor Report · Acceptance letter]

3 Jan 2023

PONE-D-22-15436R1 

Advance Care Planning (ACP) to promote receipt of value-concordant care: Results vary according to patient priorities 

Dear Dr. Prigerson:

I'm pleased to inform you that your manuscript has been deemed suitable for publication in PLOS ONE. Congratulations! Your manuscript is now with our production department. 

Kind regards, 

on behalf of

Dr. Cari Malcolm 

Academic Editor

PLOS ONE